# What Is a *2021 Reference Body*?

**DOI:** 10.3390/nu14071526

**Published:** 2022-04-06

**Authors:** Manfred J. Müller, Anja Bosy-Westphal, Wiebke Braun, Michael C. Wong, John A. Shepherd, Steven B. Heymsfield

**Affiliations:** 1Institute of Human Nutrition and Food Science, Christian-Albrechts-Universität zu Kiel, D 24105 Kiel, Germany; abosyw@nutrition.uni-kiel.de (A.B.-W.); wbraun@nutrition.uni-kiel.de (W.B.); 2University of Hawaii Cancer Center, Shepherd Res. Lab, Honolulu, HI 96816, USA; mcwong3@hawaii.edu (M.C.W.); jshepherd@cc.hawaii.edu (J.A.S.); 3Graduate Program in Nutritional Sciences, University of Hawaii at Manoa, Honolulu, HI 96822, USA; 4Pennington Biomedical Research Center, LSU System, Baton Rouge, LA 70808, USA; steven.heymsfield@pbrc.edu

**Keywords:** body composition, body shape, functional body composition, fat mass, fat free mass, skeletal muscle mass, organ masses, body protein, hydration, bone mineral content, body density, reference man, reference woman, resting energy expenditure, leptin, insulin

## Abstract

The historical *1975 Reference Man* is a ‘model’ that had been used as a basis for the calculation of radiation doses, metabolism, pharmacokinetics, sizes for organ transplantation and ergonomic optimizations in the industry, e.g., to plan dimensions of seats and other formats. The *1975 Reference Man* was not an average individual of a population; it was based on the multiple characteristics of body compositions that at that time were available, i.e., mainly from autopsy data. Faced with recent technological advances, new mathematical models and socio-demographic changes within populations characterized by an increase in elderly and overweight subjects a timely ‘state-of-the-art’ *2021 Reference Body* are needed. To perform this, in vivo human body composition data bases in Kiel, Baton Rouge, San Francisco and Honolulu were analyzed and detailed *2021 Reference Bodies*, and they were built for both sexes and two age groups (≤40 yrs and >40 yrs) at BMIs of 20, 25, 30 and 40 kg/m^2^. We have taken an integrative approach to address ‘structure–structure’ and ‘structure–function’ relationships at the whole-body level using *in depth* body composition analyses as assessed by gold standard methods, i.e., whole body Magnetic Resonance Imaging (MRI) and the 4-compartment (4C-) model (based on deuterium dilution, dual-energy X-ray absorptiometry and body densitometry). In addition, data obtained by a three-dimensional optical scanner were used to assess body shape. The future applications of the *2021 Reference Body* relate to mathematical modeling to address complex metabolic processes and pharmacokinetics using a multi-level/multi-scale approach defining health within the contexts of neurohumoral and metabolic control.

## 1. Introduction

### 1.1. A Historical View

Faced with the increasing exposure of humans to radiation due to occupational, public and medical reasons and procedures, a ‘Task Group on Reference Man’ was built in 1975 [1]. The aim of this Task Group was the calculation of justifiable external or internal radiation doses that do not cause harm in the human body for the general population. For that purpose, detailed characteristics such as physical, physiological, biological and anatomical parameters were required. A *1975 Reference Man* was created, which was mainly based on already existing *post mortem* autopsy database, it comprised heterogeneous data from different populations, races and geographical areas. As a result, the *1975 Reference Man* is a 20 to 30-year-old “Caucasian” male or female with body heights of 170 or 160cm and body weights of 70 or 58 kg resulting in BMIs of 24.2 or 22.7 kg/m^2^ and surface areas of 18,000 or 16,000 cm^2^, respectively. The *1975 Reference Man* publication was the first serious attempt to compile detailed body composition data. As such, it was a landmark publication and is still used and quoted. The *1975 Reference Man* was not a defining representative for a particular population.

In 2002, the *1975 Reference Man* had been updated and the ‘International Commission on Radiological Protection’ (ICRP) report was published [2]. This was an extended compilation of heterogeneous autopsy data sets taken from the literature or from individual authors available at the time of publication. Some of these data have never been published before and thereafter; thus, data quality is still unclear. Most of the studies included in the IRCP report had been completed more than five decades ago. Finally, in 2009, the use of the revised *1975 Reference Man* in radiation protection regulations remained pervasive in radiation protection guidance and compliance; thus, adjustments for age and sex in the case of some external exposure calculations and, thus, the end of the use of the Reference Man had been proposed [3].

### 1.2. Do We Need a 2021 Reference Body?

In 1975, a sophisticated in vivo Body Composition Analysis (BCA) was not possible, while in 2002 and 2009, suitable BCA data bases did not exist. Today, *in vivo* BCA allows addressing body compositions at the anatomical (i.e., organ/tissue level as assessed by Magnetic Resonance Imaging, MRI, and Dual-energy X-ray Absorptiometry, DXA) as well as molecular and cellular levels (as assessed by the 4 component- or 4C-Model based on hydrometry, densitometry and DXA to assess hydration and density of the body as well as bone mineral and body protein content) and, thus, characterizes masses of components, organs and tissues of the body as well as their molecular composition, i.e., their ‘qualities’. Faced with technological advances of the in vivo BCA, the development of new mathematical models and considerable socio-demographic changes (characterized by an increase in elderly and overweight subjects), it is challenging to construct a timely and ‘state-of-the-art’ *2021 Reference Body*. When compared with the *1975 Reference Man*, men and woman are now heavier, taller and have more fat mass (FM) and skeletal muscle mass (SMM), while substantial differences were also found in the composition of fat-free mass (FFM), i.e., in the masses of internal organs and skeletal muscle mass [4]. In addition, problems associated with the normalization of data (e.g., an appropriate adjustment, e.g., for height or height^2^ [5,6]) in order to compare body composition between subjects differing in their constitution and body shape have to be addressed in some detail. Thus, a future analysis is not only about the absolute masses of individual body components, organs and tissues; it is also about an understanding of the relationships between individual body components and organ and tissue masses within the dimensions of the body and metabolism.

### 1.3. Purpose of This Study

To build a *2021 Reference Body*, a human in vivo body composition data base of in-depth body composition analyses including whole body MRI together with 4C-model-data together with 3D body scans to estimate body shape will be used. To start, there is a mono-ethnic Caucasian population database of the Reference Centre for Body Composition at the Christian-Albrechts-University zu Kiel, CAU, Kiel [7,8]. In addition, a multi-ethnic database on three-dimensional optical (3DO) imaging together with a machine learning approach relating 3D body scans to body composition in humans has been built as the so-called ‘Shape Up! Adults study’ at the Pennington Biomedical Research Center, Louisiana State University, LSU, Baton Rouge, LA, USA; University of California, San Francisco, CA, USA; and the University of Hawaii Cancer Center, Honolulu, HI, USA [9,10].

## 2. Methods

This is a secondary analysis of data obtained in previous studies. The data had been obtained in different studies published previously (see refs. [7,11]), where all the study protocols and all procedures had been previously approved by the Ethics Committee of the Medical Faculty of the Christian-Albrechts-University zu Kiel, Kiel, Germany; at the Pennington Biomedical Research Center in Baton Rouge, LA, USA; at the University of California, San Francisco, CA, USA; and at the Cancer Center, University of Hawaii, in Honolulu, HI, USA. The USA studies, registered at ClinicalTrials.gov ID NCT03637855, were approved by the three center’s institutional review boards, and all participants signed an informed consent prior to evaluation.

### 2.1. Study Populations

Kiel: The characteristics of the Kiel study population have been already published before (see Table 1 in ref. [7]). The analyses of a *2021 Reference Body* were based on a total of 908 subjects of a Caucasian population (54% females, 46% males) with an age range of 18 to 86 yrs. In the total population, the range in BMI was 16.8 to 58.7 kg/m^2^ (see Figure 1 for the distributions of BMI). The population was divided according to two age groups, i.e., </> 40 yrs of age where 48 and 47% of females and males were <40 yrs, with corresponding values of 52 and 53% for age >40 yrs. The mean ages of females and males at <40 yrs were 29.2 ± 5.71 yrs and 29.3 ± 5.8 yrs. The corresponding values of females and males at >40 yrs were 61.3 ± 10.9 yrs and 61.5 ± 11.3 yrs.

Whole-body MRI data on organ and tissue masses had been obtained in a total of 818 subjects with corresponding DXA-measurements in 766 subjects. Air displacement plethysmography (ADP) had been performed in 908, while hydrometry (deuterium and sodium bromide dilution) data were obtained in 516 subjects.

Baton Rouge/Honolulu/San Francisco: A highly stratified, diverse adult (age > 18 yrs) sample (n = 570) with the approximate sex, age, weight, height and body mass index (BMI) distribution of multiethnic group of American adults (Appendix A; [9,10,11]) was evaluated as part of the Shape Up! Adults study (NIH R01 DK109008). All participants had body circumferences (mid-arm, chest, waist, hip and thigh) measured with a flexible tape, body shape evaluated with a three-dimensional optical scanner (3DO Fit3D Proscanner version 4.x; Fit3D Inc., San Mateo, CA, USA) and body composition quantified with dual-energy X-ray absorptiometry (Hologic DXA, Discovery A, Hologic, Marlborough, MA, USA). The 3DO and DXA scanners were calibrated on a regular basis according to manufacturer-specified protocols. PBRC-data had been used for circumference and surface area calculation only.

The human avatar prediction models are based on a race/ethnically mixed sample of Americans [11]. The generated avatars are, thus, racially/ethnically “generic”. We used the average values from the white Kiel sample to generate the avatars. Thus, they might differ slightly in appearance than had we developed the avatars using an all-white sample.

### 2.2. Body Composition Analyses

A detailed description of the individual methods applied for detailed BCA has been already published [12,13]. In the MRI analyses, the volumes of organs and tissues were transformed into masses by using their densities [14]. DXA was used to assess lean soft tissue and bone mineral content (BMC), while ADP had been applied to assess body volume to assess body density to calculate fat mass (FM), according to Siri’s equation. Fat free mass (FFM) is the difference between body weight and FM. Based on deuterium (D_2_O) and sodium bromide (NaBr) dilution, hydrometry characterized total body and extracellular water (TBW and ECW), respectively. The physicochemical properties of FFM were calculated assuming densities of 0.993, 1.34 and 3.038 g/mL for water, protein and minerals. Total body protein (TBPro) was calculated from the difference between FFM (as measured by ADP) and the sum of TBW and BMC. Bone mass was calculated by BMC (as measured by DXA) × 1.85. The density of FFM (DFFM) was calculated according to Guiterrez-Marin et al. [15] as follows.
DFFM (kg/L) = (TBW (L) + TBPro (kg) + BMC (kg))/(TBW (kg) × 0.99371) + TBPro (kg)/1.34 + BMC (kg)/3.038)

Imaging with 3DO was performed by a Fit3D Proscanner, which utilizes three stereo depth cameras to scan the surface of a person’s body while a platform rotates them once in a 360degree fashion. A detailed description has been previously described [9]. After 3DO data acquisition, each scan was standardized with a template as described by Allen et al. [16]. Seventy-five fiducial points from the Civilian American and European Surface Anthropometry Resource Project were placed on the meshes by trained staff using Meshlab 1.3.2 (Consiglio Nazionale delle Ricerche, Rome, Italy). Principal component analysis was then performed to build the avatar shape models [17]. The avatar images were generated using manifold regression analysis based on specified values for weight, height, age and BMI [17]. The avatar images were generated using R version 4.0.2 (R Core Teams, Vienna, Austria). The avatar circumferences and surface areas were generated using Universal Software as reported by Sobhiyeh et al. [18].

### 2.3. Assessment of Resting Energy Expenditure (REE) and Analyses of Plasma Leptin Levels

REE was measured by an open circuit indirect calorimetry after an overnight fast. Four ventilated hood systems were used (3 Vmax Spectra 29n devices (SensorMedics, Viasys Healthcare, Conshohocken, Pennsylvania, US) and one Quark RMR device, COSMED, D83413 Fridolfing, Germany) with a precision of 4.4–6.5% [13]. All REE measurements were conducted for 45 min and under steady-state conditions. Plasma leptin levels were analyzed with the use of standard laboratory techniques (as described previously in 13; i.e., DRG Leptin ELISA EIA-2395;DRG-Instruments).

### 2.4. Calculations

To build a *2021 Reference Body*, the associations between BMI and the individual body composition parameters were analyzed by Pearson and, if appropriate, by Spearman’s correlations. Separate analyses were performed according to the different age and sex groups. In addition, the individual BMI on body component (or on organ or tissue mass) regression algorithm was used to calculate the body components (and organ and tissue masses) at BMIs of 20, 25, 30 and 40 kg/m^2^ (see data presentation in Table 1). Linear and significant associations were obtained for all correlations between BMI on body component (or on organ or tissue mass). However, it is worthwhile to mention that there was a considerable variance in the explained variance of these associations. For example, in young females (i.e., <40 yrs of age) and also for males (data not shown here), the r^2^ for the BMI on the mass of subcutaneous adipose tissue (SAT) association reached 0.794. The corresponding r^2^ for the BMI on skeletal muscle mass was 0.41 and 0.03 for BMI on brain mass, respectively. Statistically, all these associations reached a significance of at least *p* < 0.05.

REE of the *2021 Reference Body* was calculated in a multiple linear regression analysis with FFM and age as determinants, resulting in a prediction algorithm.
REE (kcal/d) = 23.044 × FFM (kg) − 3.056 × age (yrs) + 53.738 (r^2^ = 0.68).

The associations between body composition parameters and metabolic (REE) and endocrine variables (plasma leptin and insulin concentrations) were analyzed by Pearson’s correlation (SPSS 28.0.10 statistical software (IBM Corporation, Armonk, NY, US).

## 3. Results

### 3.1. BMI vs. Age and BMI Distribution

To characterize the Kiel study population, the individual BMIs according to age are shown for both sexes in Figure 1. For females at ages <40 yrs and >40 yrs of age, mean body heights at BMIs of 20, 25, 30 and 40 kg/m^2^ were 167/165, 172/ 164, 168/166 and 169/168 cm with corresponding body weights of 55.8/54.5, 74.4/67.2, 84.7/82.7 and 114.2/112.9 kg respectively. The corresponding data for males were 174/179, 182/180, 182/ 180, 173/178 cm with body weights of 66.6/64.1, 82.8/81.0, 99.4/97.2 and 119.7/126.7 kg, respectively.

### 3.2. BMI vs. FM and FFM

Regression analysis was used to examine the bivariate associations between BMI and FM and FFM. In both sexes and the two age groups, there were positive associations between BMI and %FM, FM and FFM as well as the FM/FFM ratio (Figure 2, Figure 3 and Figure 4). While the relationships between BMI and FM, FFM and FM/FFM were positive and linear (Figure 3 and Figure 4), the relationship between BMI and %FM was positive but not linear (Figure 2).

### 3.3. The 2021 Reference Body at BMI 20, 25, 30, 40 kg/m^2^ and Age <40 yrs and >40 yrs

The *2021 Reference Body* was calculated from the individual associations between the BMI and the respective body component characterized by BCA. The *2021 Reference Body* is shown in Table 1. All masses and volumes of the individual body components, masses and tissues increased with BMI. When compared with females, males had greater masses of VAT; FFM; SMM; bone mass; the masses of brain, heart, liver and kidneys; the volumes of TBW and ECW; and TBProt, while FM (in kg and %bw) and SAT were higher in females. The sex differences were independent of age. When compared with the age group at <40 yrs, males and females at age >40 yrs had higher FM, SAT and VAT, while FFM, SMM, bone mass TBW and body protein were lower. For females at <40 yrs and >40 yrs of age, SMM data expressed as the percentage of body weight were 35.1/30.6, 28.6/25.2, 27.1/24.5 and 23.0/21.5% at BMIs of 20, 25, 30 and 40 kg/m^2^. In males, the corresponding values were 42.3/37.7, 38.1/34.2, 35.1/31.4 and 34.8/30.3%, respectively. In addition, the percentage muscle cells over fat cells (as expressed as the ration of SMM to FM × 100) was higher at low BMI’s compared with higher BMIs and in males compared with females. At BMIs of 20, 25, 30 and 40kg/m^2^, data for females at <40 yrs and >40 yrs of age were 123.7/114.7, 93.0/72.7, 64.2/57.1 and 44.8/42.5%, respectively. The corresponding data for males were 567.7/285.6, 184.9/142.7, 119.8/102.9 and 78.3/73.3%.

In both sex groups, there were close associations between the individual masses/volumes of body components, organs and tissues (see correlation matrix in Table 2).

D_FFM_ was calculated for males and females for the two age groups and at BMIs of 20, 25, 30 and 40 kg/m^2^. For younger and older males, D_FFM_ was 1.094/1.093, 1.093/1.089, 1.087/1.087 and 1.081/1.081. The corresponding D_FFM_-data for younger and older females were 1.096/1.090, 1.092/1.095, 1.088/1.083 and 1.081/1.079. The decreases in D_FFM_ with increasing BMI are mainly explained by the disproportional increase in TBW with increasing FFM (Table 1).

### 3.4. Associations between Whole Body and Regional Body Composition

In both sexes, there were linear and positive associations between either SAT (Figure 5A–C) or SMM (Figure 5D–F) and their respective masses at legs, arms and trunk. The slopes of the individual associations differ with steepest slopes for SAT_whole body_ vs. SAT_trunk_ and SMM_wholebody_ and SMM_legs_. As far as the regression algorithms were concerned, sex differences were seen for SAT_wholebody_ and SAT_legs_ (with a steeper slope of the regression line for females compared with males), as well as for SAT_whole_ body and SAT_trunk_ (with a steeper slope of the regression line for males compared to females).

### 3.5. Body Shape and Body Circumferences

Humanoid avatars developed using the Kiel study population [7] with mean weights, heights and ages that correspond to the specified BMIs (20, 25, 30 and 40 kg/m^2^) divided into young (age < 40 yrs) and old (age > 40 yrs) males and females are presented in Figure 6. Weight and height are matched in the young-old pairs at each BMI level. The circumferences and surface areas that correspond to these avatars are shown in Table 3. The circumferences of young (at age < 40 yrs) and old (at age > 40 yrs) male and female humanoid avatars (right) at a BMI of 25 kg/m^2^ are shown on the left in Figure 7. Greater age is accompanied by larger chest and waist circumferences and smaller hip, thigh and biceps circumferences, reflecting the redistribution of body mass with aging. These secular effects are revealed in the young–old avatar pairs. Additional circumference and surface area details are presented in Table 3.

### 3.6. Functional Body Composition (FBC)

The associations of REE on FFM, plasma leptin levels (Lep) on FM and bone mineral density (BMD) on SMM are shown in Figure 8A–C. There was a positive and linear association between FFM and REE (Figure 8A). As for the *2021 Reference Body*, REE increased with increasing BMI in males and females as well as in the two age groups of subjects (Table 4). By contrast, there was an exponential increase in Lep with increasing FM (Figure 8B). Finally, BMD increased with SMM (Figure 7C). When compared with males, females had a lower FFM with no sex differences in the REE on FFM association (Figure 8A). By contrast, there was a sex difference in the association between FM and Lep with higher Lep-for-FM-levels in females (Figure 8B). In both sexes, BMD increased with SMM with a steeper slope of the regression line in females compared with males (Figure 8C).

## 4. Discussion

### 4.1. Is the 2021 Reference Body a Step Forward?

Indirect estimates of the nutritional status, such as BMI and waist circumference, are at best crude indicators of body composition and health; they cannot be used to model metabolism, to characterize pharmacokinetics or to calculate drug and X-ray dosages and/or to optimize ergonomic calculations. Due to major technical advances in in vivo BCA together with subsequent division of masses at atomic, molecular, tissue and organ levels, a considerable heterogeneity of the individual body components became evident [19,20]. As far as the functional impact of individual body components on metabolism and cardiometabolic risks is concerned, diverse inter-relations between body components, organs, tissues and metabolism as well as its disturbances have been demonstrated (see discussion in [7,12,19]). Combining the organ/tissue level (as assessed by MRI and DXA) with the cellular and molecular model (as characterized by the 4C-model) adds physical characteristics (e.g., hydration and density) of organs and tissues and, thus, characterizes the composition of individual body components (i.e., their ‘quality’) where both masses as well as their ‘qualities’ impact metabolism [21]. Finally, combining in vivo BCA data with the avatar shape models can be used to characterize specific phenotypes, e.g., in patients with obesity [11,17].

After a period of 45 yrs, the *2021 Reference Body* is a timely application of the concept of the 1975 Reference Man. The *2021 Reference Body* is based on data obtained by *state-of-the-art* methods and in vivo technologies to assess body composition at the anatomical and molecular levels. In addition, the *2021 Reference Body* refers to the different BMI categories and, thus, allows a differentiated view at different nutritional states in younger and older adults (Table 1 and Table 2; Figure 2, Figure 3, Figure 4, Figure 5 and Figure 6), taking into account specific ‘structure–structure’ (Table 2) and ‘structure–function relationships’ (Figure 8; Table 4). Furthermore, the *2021 Reference Body* readdresses a fundamental question of human biology: i.e., the association between body shape and body composition (Table 3; Figure 7). Finally, the present analysis considered the relationships between FM and FFM (Figure 4) and between individual body components of the entire body (i.e., SAT and SMM) and their masses within different body regions (Figure 5). While referring to the concept of ‘Functional Body Composition’ (FBC), it relates individual body components, organs and tissues to some of their related functions and, thus, adds to an understanding of the meaning and the impact of the *2021 Reference Body*.

### 4.2. Should the 2021 Reference Body Replace the 1975 Reference Man?

When compared with the *2021 Reference Body*, our first evaluation of the *1975 Reference* Man had served as a basis for the present project [4]. The study population back then was relatively small and had a limited age range. Furthermore, in our previous study [4], MRI-derived skeletal muscle mass, the molecular composition of the components of FFM and functional characteristics (e.g., on REE and plasma levels of leptin and insulin) were not included. Finally, a normative approach closely corresponding to the *1975 Reference Man* had only been used.

The *1975 Reference Man* is a comprehensive compilation of anatomical and physiological human data [1]. When compared with the *2021 Reference Body*, the *1975 Reference Man* has had a much bigger focus. The report contains far reaching anatomical data, e.g., on the brain and nervous system; skin; bone marrow (including the hematopoietic and lymphatic system); the gastrointestinal tract (including the surface areas of different parts of the intestine); the respiratory and the urogenital systems; and the endocrine and the cutaneous system. In addition, there is an extensive list of 51 different element contents in the body that are used to model values for the daily balance of elements. Finally, the *1975 Reference Man* is about children as well as adults.

While the *1975 Reference Man* had been built from data taken from some untraceable sources, the *2021 Reference Body* has been measured in two well-defined study populations. Since in the present study, the observed densities of FFM resemble those reported for other populations (e.g., published in a standard textbook of body composition [22]), this may support the generalizability of the present data. However, it is obvious that the data available for characterizing the *2021 Reference Body* also reflect more recent BCA research activities, which had mainly addressed issues of energy balance, insulin resistance, cardiometabolic risk assessment and modelling of metabolic and physical functioning related to energy balance and its disturbance. Thus, the *2021 Reference Body* results from research activity address the structural determinants of metabolic functioning rather than calculating radiation doses (as has been the aim of the authors of the *1975 Reference Man*). It follows that the *2021 Reference Body* complements rather than replaces the *1975 Reference Man*. Comparing the two references also reflects changes in the scope and concepts of research on body composition.

### 4.3. Suitable Applications of the 2021 Reference Body

Mathematical models of human metabolism and changes in body weight (and body composition use multi-level/multi-scale analyses that take into account linear and non-linear relationships between individual body components, organs and tissues and between body components and metabolism (including physical activity and metabolic adaptations as well as the determinants of metabolism and related functions [23,24,25,26,27,28,29];). The applications of mathematical models of metabolism to weight changes suggest that body weight control is based on the relationships between organs, tissues and individual body components rather than on their absolute masses [30,31]. This advanced understanding results in a dynamic rather than a static prediction model of weight changes in response to energy balance interventions [31].

In addition, the *2021 Reference Body* together with the concept of FBC can serve as a basis for defining phenotypes related to specific metabolic and endocrine traits. Suitable applications of this concept are interpretations of body functions (e.g., energy intake and energy expenditure) and their disturbances in the context of body components and vice versa and understanding of the meaning of individual body components in the context of their functional consequences [12,31]. In a virtual model, masses and the regression coefficients relating REE to FFM (or plasma leptin levels to FM or BMD to SMM, see Figure 8) can be used as model parameters.

Physiological-based pharmacokinetic (PBPK) modeling offers a mechanism-based approach to pharmacokinetic modeling, where drug concentration-time profiles are described in the respective organs and body fluids considering the physiological system (=human body and body composition at the anatomical and molecular level) and the physicochemical properties of the drug [32,33]. To build a proper PBPK model, two different sets of parameters are required.

First, drug-dependent physicochemical parameters include lipophilicity, protein binding, molecular weight, solubility and the acid-dissociation constant.

Second, system dependent parameters include drug-independent physiological parameters such as organ volumes, blood flow rates, effective surface areas for permeation processes and pH values.

The basic principle of PBPK modeling is to segment the mammalian body into physiological relevant compartments and develop mathematical expressions of physical and physiological processes for each compartment describing the fate of the compound within the respective compartment. Exceedingly, easily accessible compartments such as plasma, saliva and urine reflect major organs such as liver, heart, lung, muscle, brain, etc., and each compartment is further subdivided into sub-compartments (i.e., interstitial, intra- and subcellular space). This model can be used to make predictions of pharmacokinetic behavior in specific virtual individuals or populations using realistic physiological, anatomical and molecular properties. Based on the data of the *2021 Reference Body*, a virtual population simulator can be generated using non-linear mixed effects modeling technique, which allows the estimation of population means for model parameters and the quantification of inter-individual variability and residual (unexplained) variability.

### 4.4. Future Directions of the 2021 Reference Body

Although the *1975 Reference Man* as well as the *2021 Reference Body* are based on data obtained in apparently healthy subjects, the data are polluted by health risks common in populations living in Western societies; e.g., some of apparently healthy subjects are at an increased risk to develop Non-Communicable Diseases such as obesity, cardiovascular diseases, diabetes mellitus and certain cancers, which were unrecognized at the time of our examination. Although a global consideration of this issue is beyond the scope of this paper, that issue has to become a component of documentation for the upcoming versions of the *Reference Man*.

The two data bases used to calculate and describe the *1975 Reference Man* and the *2021 Reference Body* did not include certain populations, e.g., subjects at advanced ages or an Asian population. It is worthwhile to note that the *1975 Reference Man* and the *2021 Reference Body* publication were serious attempts to compile detailed body composition data for the purpose of modelling and calculations. By contrast, the data do not define a true representative for a particular population.

Presently, the *2021 Reference Body* is merely descriptive. As a next step to a future Reference Body, a mathematical approach to address the hierarchy of organs, tissues and body components using ‘structure–structure’ as well as ‘structure–function’ relationships will allow proceeding beyond the merely descriptive level [31]. For example, during caloric restriction, brain glucose oxidation is maintained at the cost of hepatic energy expenditure due to increases in ATP-consuming gluconeogenesis. In fact, energy expenditure due to hepatic gluconeogenesis may be as high as almost one-third of REE [34]. With fasting, hepatic energy oxygen consumption may increase with a concomitant decrease in whole-body REE [13,35]. Thus, in this situation, the maintenance of brain mass, its metabolism and energy needs to determine both hepatic and whole-body energy expenditure, suggesting that, in specific metabolic situations, a hierarchy of organs and tissues must be implemented in a future Reference Body, which will add to an analytical level of the present descriptive *2021 Reference Body*. Such hierarchical reference data shall be used as input for various continuative models, e.g., future dynamic models of weight changes [30].

Further extending today’s BCA to the intra-cellular level and, thus, remodeling the molecular level provides another perspective for the future Reference Body. By analyzing a mass-spectrometry-based reference lipidome in human white adipose tissue biopsies, over 1600 and 700 lipid species can be identified qualitatively and quantitatively [35,36]. The molecular mapping of the lipidome in normal weight subjects and patients with obesity showed considerable inter-individual diversity (e.g., PUFA-containing triglycerides associated with larger lipid droplets and atypical ceramides were upregulated in obese SAT). In addition, plasmalogen phospholipid signatures differed between adipocytes from SAT and from visceral adipose tissue (VAT). Thus, assessing the masses of SAT and VAT together with the measurement of the tissue-specific lipidome defined obesity-mediated lipid alterations, which may provide insight into the functional aspects of adipose tissues as well as the etiology of associated diseases. To address adipose tissue dysfunction, mass spectrometry-based imaging approaches will add to the lipidome signatures of SAT and VAT. However, implementing human SAT and VAT reference lipidome characteristics into a future Reference Body provides a functional evaluation of individual body components, organs and tissues, which may serve to be powerful in a virtual population simulator of metabolism and health risks. Interestingly, in addition to this so-called ‘AdipoAtlas’ (which is freely available for all researchers, [37]), there is a ‘Human Proteome Atlas project’ with expression levels of mRNA and proteins reported for 44 healthy human tissues [38]. These data bases can be used to characterize functional aspects of tissues, cell type-specific signatures, future drug targets and potential biomarkers.

To summarize, the *2021 Reference Body* results from a robust body composition data base. Combined with functional and intracellular characteristics, it has a lot of potential for further developments and applications. The *2021 Reference Body* provides a sound basis for mathematical modeling to address complex metabolic processes, pharmacokinetics and ergonomic optimizations.

## Figures and Tables

**Figure 1 nutrients-14-01526-f001:**
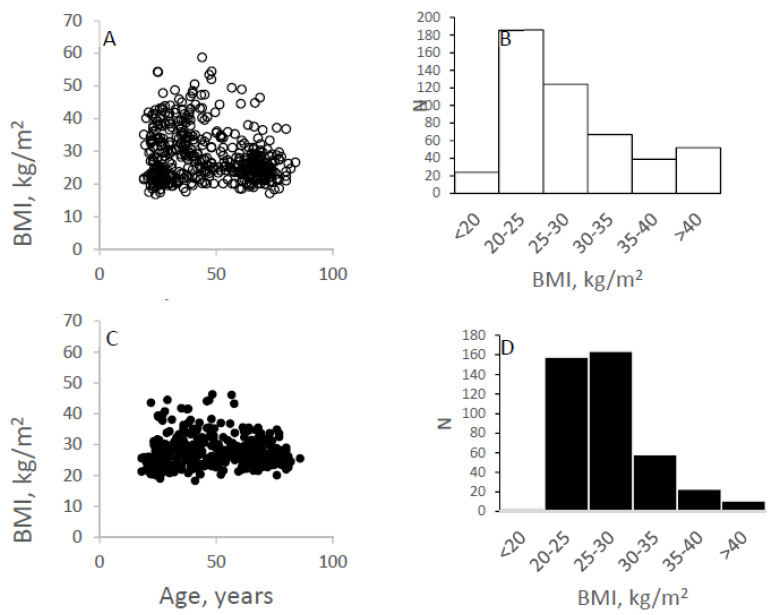
BMI with age in females (**A**) and males (**C**) and the distribution of BMI groups according to BMI categories (<20, 20–25, 25–30, 30–35, 35–40 and >40 kg/m^2^) in females (**B**) and males (**D**), respectively, in the ‘Kiel study population’: n = 908 subjects; 54% females and 46% males; age range of 18 to 86 yrs.

**Figure 2 nutrients-14-01526-f002:**
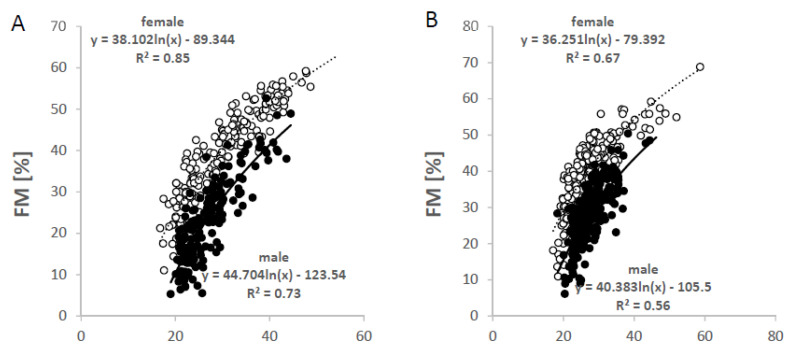
Non-linear plot of the association between BMI and percentage fat mass (%FM, as estimated by ADP) in females (open symbols) and males (closed symbols) at age ≤40 yrs (**A**) and >40 yrs (**B**) (‘Kiel study population’, n = 908 subjects; 54% females and 46% males; age range of 18 to 86 yrs). As for the association between %FM and BMI, BMI data were transformed to the natural logarithm scale to minimize non-linearity.

**Figure 3 nutrients-14-01526-f003:**
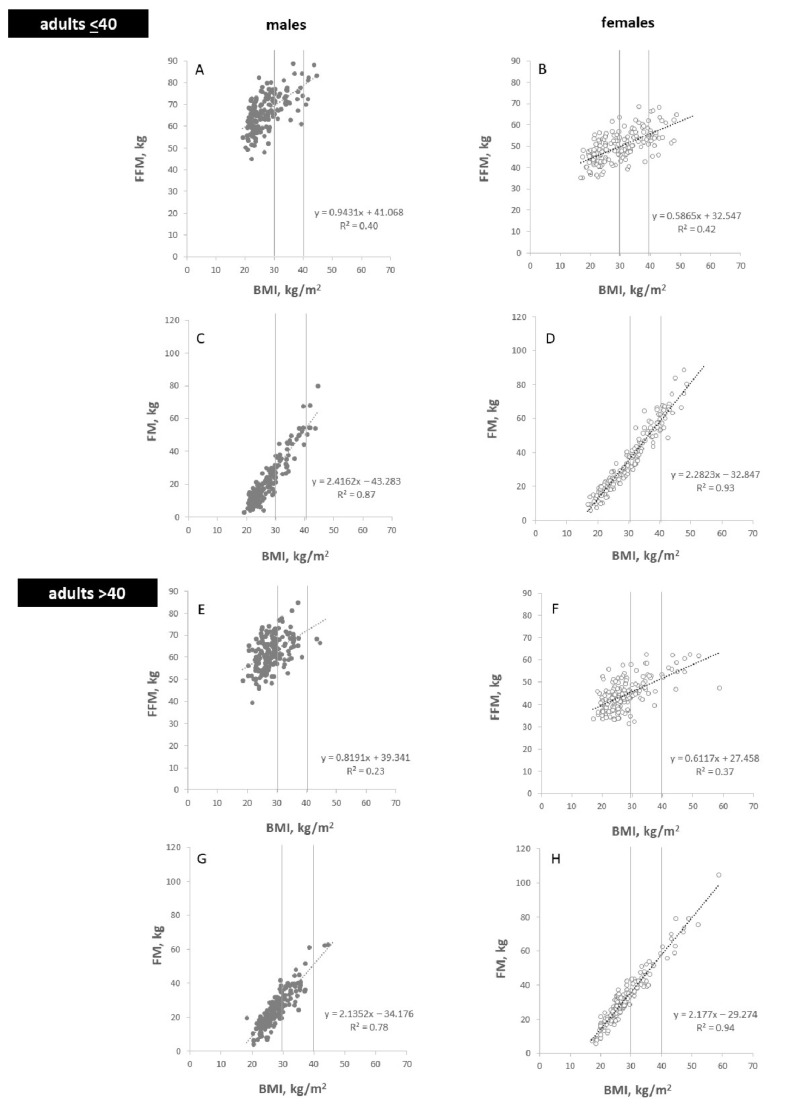
Associations between BMI and either FFM (**A**,**B**,**E**,**F**) and FM (**C**,**D**,**G**,**H**) in males (**A**,**C**,**E**,**G**) and females (**B**,**D**,**F**,**H**) in adults at age ≤40 yrs (**A**–**D**) and >40 yrs (**E**–**H**). Characteristics of the ‘Kiel study population’: n = 908 subjects; 54% females and 46% males; age range of 18 to 86 yrs with 52% females and 52% males at ≤40 yrs of age.

**Figure 4 nutrients-14-01526-f004:**
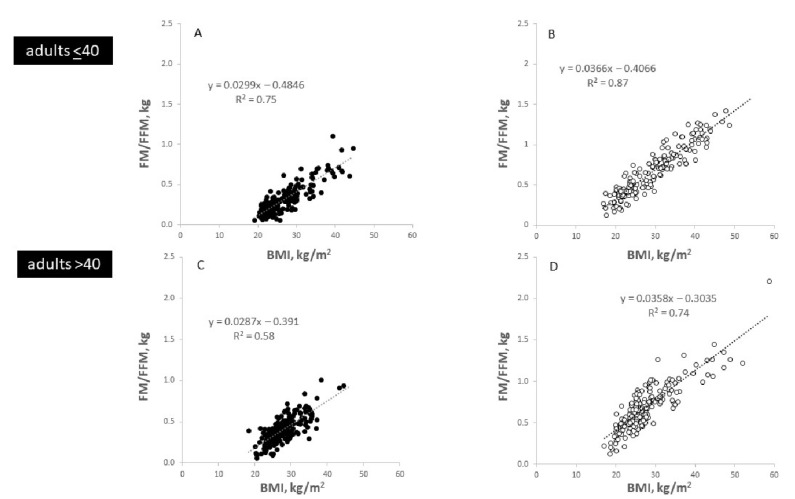
Associations between BMI and the FM/FFM-ratio in males (**A**,**C**) and females (**B**,**D**) in adults at age <40 yrs (**A**,**B**) and >40 yrs (**C**,**D**). Characteristics of the ‘Kiel study population’: n = 908 subjects; 54% females and 46% males; age range of 18 to 86 yrs with 52% females and 52% males at ≤40 yrs of age.

**Figure 5 nutrients-14-01526-f005:**
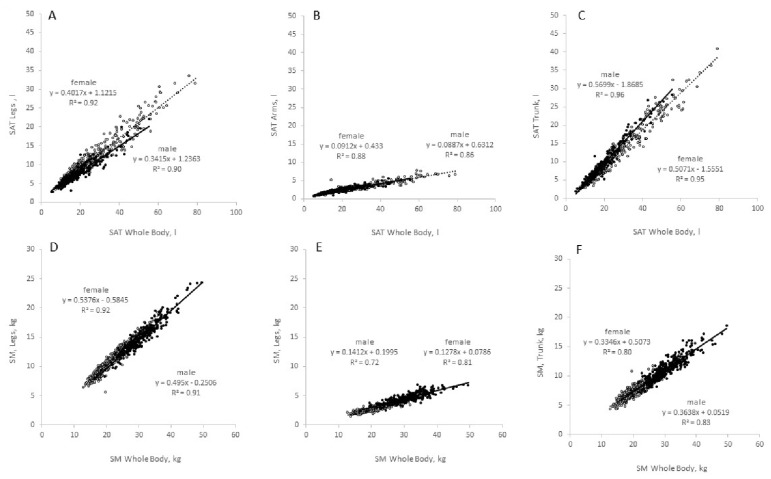
Associations between whole body and regional body components. Upper panel: Whole body subcutaneous adipose tissue (SAT) vs. SAT at legs (**A**), arms (**B**) and trunk (**C**). Lower panel: Whole body skeletal muscle mass (SMM) vs. SMM of legs (**D**), arms (**E**) and trunk (**F**). Open symbols, females, n = 355 for SAT and 349 for SMM; closed symbols, males, n = 321 for SAT and 318 for SMM. Subpopulations were taken from the ‘Kiel study population’ (see Methods).

**Figure 6 nutrients-14-01526-f006:**
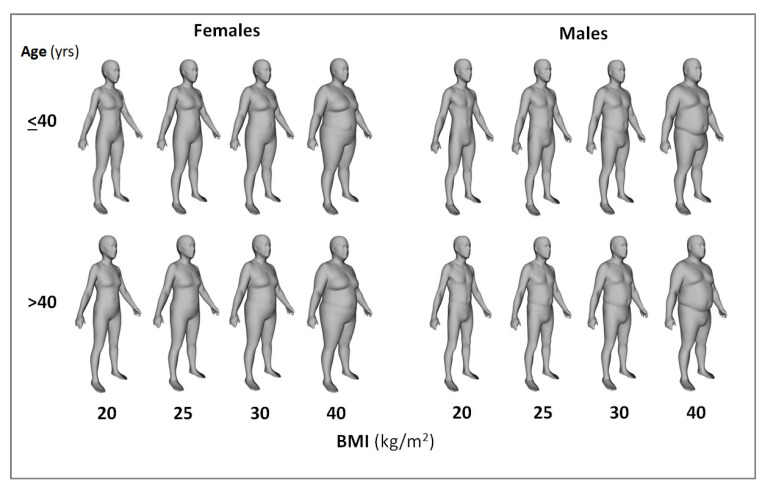
Humanoid avatars that correspond to the four specified BMIs divided into young and old males and females. Corresponding circumferences and surface areas of these avatars are shown in Table 3. For details of the ‘Shape Up! Adults study’ sample, see Appendix A).

**Figure 7 nutrients-14-01526-f007:**
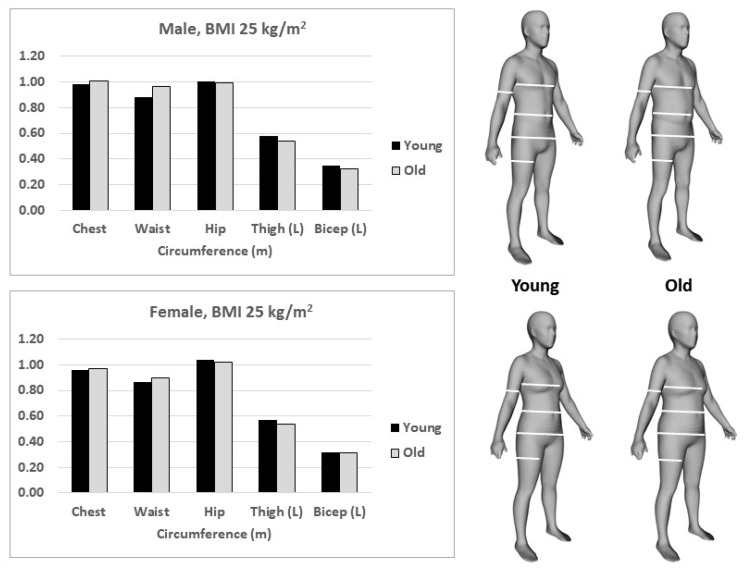
Circumferences of young (≤age 40 yrs) and old (>age 40 yrs) male and female humanoid avatars (right) at a BMI of 25 kg/m^2^ are shown on the left. For details of the ‘Shape Up! Adults study’ sample, see Appendix A).

**Figure 8 nutrients-14-01526-f008:**
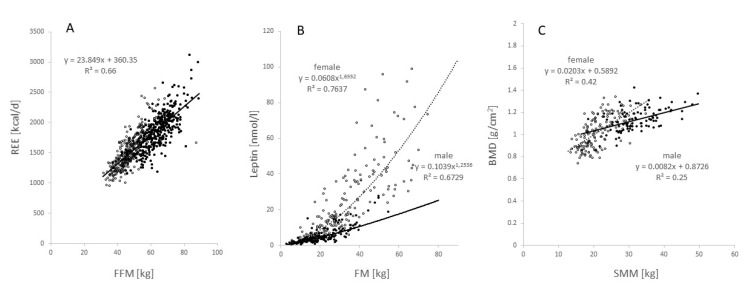
Functional body composition: (**A**) REE on fat free mass (FFM; n = 364 females and 320 males); (**B**) plasma leptin levels (Lep) on fat mass (FM; n = 178 females and 174 males); and (**C**) bone mineral density (BMD) on skeletal muscle mass (SMM; n = 149 females with 33.5% at age >40 yrs and 93 males with 22.5% at age >40 yrs). Subpopulations were taken from the ‘Kiel study population’ (see Methods).

**Table 1 nutrients-14-01526-t001:** Detailed body composition at the anatomical and molecular level of the *2021 Reference Body* for males and females according to BMI categories and age of <40 yrs and >40 yrs. Since there were no significant associations between BMI and the masses of brain and heart within the individual age groups in males, the respective reference values are provided for the entire population. See Methods or details of the ‘Kiel study population’.

BMI, kg/m^2^	*Age*	Male				Female			
		20	25	30	40	20	25	30	40
FM, kg; %	*>40 yrs*	8.47	19.44	30.40	52.33	14.69	25.47	36.24	57.79
		15.48	24.49	31.85	43.47	29.21	37.30	43.90	54.33
	≤*40 yrs*	4.95	17.04	29.13	53.31	24.32	12.90	35.74	58.58
		10.38	20.36	28.51	41.37	24.80	33.30	40.25	51.21
SAT, L	*>40 yrs*	10.22	15.78	21.35	32.48	16.90	22.35	27.81	38.72
	≤*40 yrs*	7.26	14.76	22.27	37.28	14.94	22.80	30.66	46.38
VAT, L	*>40 yrs*	1.68	3.27	4.86	8.04	1.08	1.96	2.85	4.62
	≤*40 yrs*	0.74	2.22	3.71	6.68	0.89	1.43	1.20	3.10
FFM, kg	*>40 yrs*	55.38	59.63	63.87	72.36	38.62	41.76	44.91	51.20
	≤*40 yrs*	60.05	64.74	69.43	78.81	43.98	47.16	50.34	56.70
SM, kg	*>40 yrs*	24.19	27.74	31.28	38.37	16.85	18.78	20.72	24.59
	≤*40 yrs*	28.10	31.51	34.91	41.72	19.64	21.30	22.96	26.28
Bone, kg	*>40 yrs*	4.67	4.77	4.87	5.07	3.43	3.65	3.86	4.33
	≤*40 yrs*	5.21	5.32	5.43	5.65	4.20	4.33	4.46	4.72
Brain, kg	*>40 yrs*	1.527	1.547	1.567	1.607	1.319	1.339	1.359	1.399
	≤*40 yrs*					1.385	1.395	1.405	1.425
Heart, kg	*>40 yrs*	0.344	0.359	0.374	0.404	0.291	0.281	0.271	0.251
	≤*40 yrs*					0.261	0.271	0.281	0.301
Liver, kg	*>40 yrs*	1.225	1.545	1.865	2.505	1.022	1.272	1.522	2.022
	≤*40 yrs*	1.439	1.714	1.989	2.539	1.286	1.451	1.616	1.946
Kidneys, kg	*>40 yrs*	0.246	0.276	0.306	0.366	0.195	0.220	0.245	0.295
	≤*40 yrs*	0.234	0.274	0.314	0.394	0.218	0.243	0.268	0.318
TBW, L	*>40 yrs*	40.29	44.86	49.43	58.57	29.80	33.20	36.60	43.41
	≤*40 yrs*	40.95	46.96	52.98	65.01	31.10	35.29	39.50	47.92
TBW/FFM, L/kg	*>40 yrs*	0.72	0.74	0.76	0.80	0.74	0.76	0.78	0.82
	≤*40 yrs*	0.70	0.73	0.75	0.80	0.70	0.73	0.76	0.82
ECW, L	*>40 yrs*	16.38	18.01	19.66	22.94	13.10	14.17	15.24	17.38
	≤*40 yrs*	16.77	17.97	19.17	21.57	13.10	14.37	15.68	18.30
Protein, kg	*>40 yrs*	13.12	14.41	15.70	18.26	9.11	9.66	10.21	11.31
	≤*40 yrs*	13.23	14.83	16.42	19.61	9.81	10.78	11.75	13.69

**Table 2 nutrients-14-01526-t002:** Cross-correlation matrix of body components, organs tissues in females (italic) and males (normal) of the ‘Kiel study population’ (see Methods for further details).

	FM, kg	SAT, L	VAT, kg	FFM, kg	SM, kg	Bone, kg	Brain, kg	Heart, kg	Liver, kg	Kidney, kg	TBW, L	ECW, L	Protein, kg
FM, kg		*0.978 ****	*0.541 ****	*0.581 ****	*0.637 ****	*0.267 ****	*0.153 ***	*0.136 **	*0.665 ****	*0.506 ****	*0.730 ****	*0.640 ****	*0.357 ***
SAT, L	0.936 ***		*0.529 ****	*0.636 ****	*0.652 ****	*0.392 ****	*0.232 ****	*0.343 ****	*0.497 ****	*0.575 ****	*0.756 ****	*0.616 ****	*0.468 ****
VAT, kg	0.682 ***	0.591 ***		*0.340 ****	*0.448 ****	*0.217 ****	*0.001* (*NS*)	*0.310 ****	*0.317 ****	*0.301 ****	*0.252 ****	*0.204 ****	*0.040* (*NS*)
FFM, kg	0.300 ***	0.447 ***	0.228 ***		*0.869 ****	*0.785 ****	*0.384 ****	*0.233 ****	*0.700 ****	*0.522 ****	*0.885 ****	*0.777 ****	*0.882 ****
SM, kg	0.442 ***	0.506 ***	0.313 ***	0.874 ***		*0.684 ****	*0.292 ****	*0.311 ****	*0.796 ****	*0.611 ****	*0.882 ****	*0.782 ****	*0.779 ****
Bone, kg	−0.041 (NS)	0.066 (NS)	0.057 (NS)	0.764 ***	0.534 ***		*0.273 ****	*0.233 ****	*0.452 ****	*0.120 **	*0.554 ****	*0.479 ****	*0.573 ****
Brain, kg	0.104 (NS)	0.283 ***	0.246 ***	0.300 ***	*0.247 ****	*0.187 ***		*-0.033* (*NS*)	*0.312 ****	*0.347 ****	*0.361 ****	*0.229 ****	*0.304 **
Heart, kg	0.133 *	0.229 ***	0.255 ***	0.179 **	0.024 (NS)	0.302 ***	0.034 (NS)		*-0.066* (*NS*)	*0.209 ****	*0.445 ****	*0.352 ****	*0.261 **
Liver, kg	0.554 ***	0.556 **	0.410 ***	0.678 ***	0.685 ***	0.349 ***	0.308 ***	0.071 (NS)		*0.381 ****	*0.760 ****	*0.719 ****	*0.604 ****
Kidney, kg	0.326 ***	0.498 ***	0.430 ***	0.395 ***	0.376 ***	0.139 *	0.420 ***	0.152 *	0.546 ***		*0.602 ****	*0.449 ****	*0.503 ****
TBW, L	0.496 ***	0.571 ***	0.258 ***	0.857 ***	0.898 ***	0.555 ***	0.278 ***	0.079 (NS)	0.691 ***	0.374 ***		*0.806 ****	*0.775 ****
ECW, L	0.346 ***	0.357 ***	0.147 *	0.685 ***	0.702 ***	0.506 ***	0.026 (NS)	0.282 ***	0.584 ***	0.281 ***	0.758 ***		*0.786 ****
Protein, kg	0.232 *	0.316*	0.211 *	0.774 ***	0.676 ***	0.545 ***	0.228 *	0.169 (NS)	0.530 ***	0.228 *	0.658 ***	0.601 ***	

* *p* < 0.05; ** *p* < 0.01; *** *p* < 0.001; NS, not significant. ECW, extracellular water; FFM, fat-free mass; FM, fat mass; SAT, subcutaneous adipose tissue; SM, skeletal muscle; TBW, total body water; VAT, visceral adipose tissue.

**Table 3 nutrients-14-01526-t003:** Circumferences and Surface areas of the *2021 Reference Body* for males and females according to BMI categories and age of <40 and >40 yrs. See Supplementary Material for details for the Shape Up! Adults study sample, Appendix A).

BMI, kg/m^2^	*Age*	Male				Female			
		20	25	30	40	20	25	30	40
Circumferences									
Rt Bicep, m	*>40 yrs*	0.298	0.335	0.370	0.425	0.269	0.309	0.342	0.427
	*≤40 yrs*	0.287	0.319	0.355	0.425	0.267	0.302	0.337	0.446
Lt Bicep,m	*>40 yrs*	0.309	0.345	0.378	0.439	0.277	0.316	0.351	0.421
	*≤40 yrs*	0.296	0.325	0.359	0.426	0.276	0.311	0.347	0.425
Chest, m	*>40 yrs*	0.882	0.983	1.059	1.203	0.870	0.959	1.038	1.178
	*≤40 yrs*	0.921	1.005	1.083	1.246	0.885	0.970	1.053	1.163
Waist, m	*>40 yrs*	0.742	0.885	1.008	1.213	0.756	0.864	0.968	1.217
	*≤40 yrs*	0.823	0.963	1.073	1.270	0.788	0.899	1.015	1.269
Hip, m	*>40 yrs*	0.922	1.006	1.079	1.212	0.956	1.037	1.106	1.285
	*≤40 yrs*	0.919	0.997	1.078	1.224	0.935	1.019	1.119	1.279
Rt Thigh, m	*>40 yrs*	0.508	0.579	0.631	0.698	0.500	0.569	0.622	0.724
	*≤40 yrs*	0.489	0.538	0.590	0.673	0.491	0.539	0.602	0.694
Lt Thigh, m	*>40 yrs*	0.512	0.580	0.629	0.701	0.500	0.571	0.619	0.716
	*≤40 yrs*	0.491	0.537	0.588	0.674	0.492	0.539	0.601	0.689
Surface Areas									
Head, m^2^	*>40 yrs*	0.182	0.198	0.208	0.219	0.170	0.179	0.183	0.199
	*≤40 yrs*	0.182	0.190	0.202	0.223	0.170	0.176	0.184	0.198
Rt Arm, m^2^	*>40 yrs*	0.172	0.196	0.208	0.219	0.149	0.165	0.171	0.199
	*≤40 yrs*	0.177	0.188	0.202	0.224	0.145	0.153	0.167	0.204
Lt Arm, m^2^	*>40 yrs*	0.178	0.200	0.211	0.219	0.151	0.168	0.173	0.198
	*≤40 yrs*	0.179	0.190	0.201	0.223	0.148	0.155	0.169	0.198
Trunk, m^2^	*>40 yrs*	0.479	0.554	0.624	0.736	0.460	0.523	0.550	0.694
	*≤40rsy*	0.511	0.562	0.643	0.807	0.454	0.502	0.550	0.726
Legs, m^2^	*>40 yrs*	0.686	0.801	0.833	0.815	0.652	0.732	0.772	0.845
	*≤40 yrs*	0.701	0.762	0.785	0.812	0.637	0.675	0.748	0.791
Total, m^2^	*>40rsy*	1.680	1.928	2.062	2.182	1.566	1.748	1.829	2.109
	*≤40 yrs*	1.731	1.874	2.011	2.262	1.538	1.644	1.797	2.090

**Table 4 nutrients-14-01526-t004:** Resting energy expenditure of the *2021 Reference Body* for males and females according to BMI categories, sex and age (of *<40 yrs* and *>40 yrs*). See Methods or details for the ‘Kiel study population’ and the calculation of the data.

		Males		Females	
	Age	*≤40 yrs*	*>40 yrs*	*≤40 yrs*	*>40 yrs*
BMI, kg/m^2^					
20		1828 kcal/d	1622 kcal/d	1458 kcal/d	1236 kcal/d
25		1936 kcal/d	1720 kcal/d	1531 kcal/d	1309 kcal/d
30		2044 kcal/d	1818 kcal/d	1605 kcal/d	1381 kcal/d
40		2260 kcal/d	2013 kcal/d	1751 kcal/d	1526 kcal/d

## Data Availability

Unfortunately, data described in the manuscript, code book and analytic code will not be made available because of the strict rules of the German Data Protection Law. In addition, completion of the scientific work on these data has not been accomplished yet.

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
