# Peer review of "What Is a 2021 Reference Body?"

_nutrients, 2022, doi:10.3390/nu14071526_

Round 1
Reviewer 1 Report
The authors are to be commended upon the scope and undertaking of this study. The availability of their databases placed them in a unique position to be able to develop the new Reference Body. The manuscript is clearly written and accessible to the reader. I have only a few minor points for consideration.
Title Why not Reference Body, i.e. capitalize Body seems more logical?
Line 104 and throughout m/s. Why split at 40 years? I have no problem with this but the choice of 40 rather than an other age needs to be justified.
Lines 100 to 113. I appreciate that details of the study population are to be found elsewhere but it is not clear to me here whether the study sample size used here was 908, 766, 864 or 516 or all depending upon the parameter. Also it is not clear whether they were all the same subjects (i.e. 908 but with missing data for some parameters). Can this be made clear please?
Line 127/8. This seems to be inconsistent: a potential difference if an all white sample had been used but the previous sentence states that a "white Kiel sample" was used.
Line 135 D2O - subscript.
Line 137 "...(TBW and ECW) respectively"
Line 171 and elsewhere throughout the manuscript. Do you mean Table not Tab.? Table is also used elsewhere. Please be consistent.
Line 175 r2 should be r2 also elsewhere please check m/s.
Results - Figures. These refer to 908 sample size - see point above.
Lines 253-258. These density values are very close to those previously published, e.g. Wang et al. 2002). It might be worth noting this agreement. Equally, there is little comment on whether other data are concordant with previously published information. Particularly, if from other populations this would potentially support the generalizability of the present data.
Discussion. This reads well and covers the importance of the study without being speculative. It might be appropriate here to make comment about the agreement between the present data and pre-existing data as noted above.
Line 356 "..previous study". please provide the source citation.
Line 370. "undeniable sources" Undeniable does not seem to me to be the most appropriate description here - do you mean reputable?
Line 496. I appreciate that the authors wish to retain ownership of the database for their future analysis but is disappointing that this valuable database cannot be made more widely available - perhaps upon reasonable application/collaboration?
Author Response
Thank you for discussing with us, and thanks for your advices, which we found helpful.
ln 104, we have spirited our population at age 40yrs according to the median of age of the total study population, i.e., this was near to age 40yrs. We first thought about working on age groups of 20-60yrs and >60yrs (as we had done before in an descriptive paper, see J Gerontol A Biol Sci Med Sci, 2016, Vol. 71, No. 7, 941–946. There we had studied detailed BC of males and females in age groups of 18-30, 30-39, 40-49, 50-59 and >60yrs. However, that approach resulted in smaller subgroups which limits a more general view on a Reference body.
ln100-113, the reviewer is right pointing out to the different no of subjects. We thus took the opportunity to go back to our data base. The no of 908 relates to ADP data, this has been corrected in the text. There were different no for the different BCA methods applied but they were all the same subjects.
ln127/8, the analyses of the Kiel data has been done first. We have then tried to make the US data comparable to our data referring to the same BMI in the two populations.
ln135, has been done.
ln137, corrected.
ln171, Tables and Figures have replaced Tab. and Fig. throughout the ms.
ln175, corrected.
ln253-258, thanks for that argument which is now used in the discussion. We have also added a new reference, i.e., 22.
Discussion, see our previous answer to your comment related to ln253-258.
ln356, added, i.e., ref.4.
ln370, we have used 'untraceable sources', i.e., we could not find that sources. By contrast, we felt that using 'reputable' is a too hard argument.
ln496, in general we agree with your point. However, presently, we are reserved. This is because of some more or less bad experiences. In the past we have already given our data base to some authors who had requested it. Unfortunately we never heard again from these authors. This was true even after further requests to them. Call me old-fashioned, but I feel that these data (when used properly) should serve as a joint basis of discussion to the benefit of a better understanding of all of us (including ourselves as the primary authors.

Reviewer 2 Report
Dear authors,
I should first like to congratulate you for the interesting, elabirate and useful work for scientific aims.
I think that your data are very involved and consistents . This is me expressing concern about the difficult for me to understand some results; i suggest that you should improvd for an easier and a better understanding of the readers the tables and figures' formatting.
You have to best format the tables , about the figures you have to make these available to a larger sizes.
Regarding the discussione , I recommend you to add additional information about the suitable applications and future aims of these endpoints/data results.
Author Response
Thanks for your responses.
By some re-formulations in the discussion we have tried to make our ms easier to understand.
Re-formatting of figures is also been tried, i.e., for Figure 2. However, this is difficult to do because of the format of the journal itself. This is not to evade but looking at the present formats of figures in our top journals (like Nature, Cell etc), it becomes obvious that publishers and editors have changed the format of figures to the format of stamps which are not readable anymore. Thus, we feel that your point is also up to the publisher. Anyhow, we have done our best to improve the readability of the ms (including the Figures).
We have already provided some ideas about future applications of the Refbody. Since we are working in the area of physiology we do not want to go beyond our horizons, e.g., to add some more ideas about suitable applications in industry design etc.
